Skimming for barcodes: rapid production of mitochondrial genome and nuclear ribosomal repeat reference markers through shallow shotgun sequencing

Hoban Mykle L. mhoban@hawaii.edu mh@myklehoban.com 1
Whitney Jonathan 2
Collins Allen G. 3
Meyer Christopher 4
Murphy Katherine R. 5
Reft Abigail J. 3
Bemis Katherine E. 3
1 Hawai‘i Institute of Marine Biology, University of Hawai‘i at Mānoa , Kāne‘ohe , Hawai‘i , United States of America
2 Pacific Islands Fisheries Science Center, National Oceanic and Atmospheric Administration , Honolulu , Hawai‘i , United States of America
3 NOAA National Systematics Laboratory, Natural Museum of Natural History, Smithsonian Institution , Washington , D.C. , United States of America
4 Department of Invertebrate Zoology, National Museum of Natural History, Smithsonian Institution , Washington , D.C. , United States of America
5 Laboratories of Analytical Biology, National Museum of Natural History, Smithsonian Institution , Washington , D.C. , United States of America
Pochon Xavier
Electronic publication date: 2022 Aug 5
Publication date: 2022
Volume: 10
Electronic Location ID: e13790
Received 2022 May 3; Accepted 2022 Jul 5
Copyright: ©2022 Hoban et al.
Copyright year: 2022
Copyright holder: Hoban et al.
License: This is an open access article distributed under the terms of the Creative Commons Attribution License, which permits unrestricted use, distribution, reproduction and adaptation in any medium and for any purpose provided that it is properly attributed. For attribution, the original author(s), title, publication source (PeerJ) and either DOI or URL of the article must be cited.
License URL: https://creativecommons.org/licenses/by/4.0/

Keywords: DNA barcoding, Genome skimming, Mitochondrial genomes, Fishes, Metabarcoding, Collections, Museum

Funding: NOAA’s Office of Science and Technology NOAA’s Pacific Islands Fisheries Science Center The Cooperative Institute for Marine and Atmospheric Research NOAA’s West Hawai‘i Integrated Ecosystem Assessment Program NMNH Laboratories of Analytical Biology The authors received funding support from NOAA’s Office of Science and Technology, NOAA’s Pacific Islands Fisheries Science Center, The Cooperative Institute for Marine and Atmospheric Research, NOAA’s West Hawai‘i Integrated Ecosystem Assessment Program, and NMNH Laboratories of Analytical Biology. The funders had no role in study design, data collection and analysis, decision to publish, or preparation of the manuscript.

==============================
DNA barcoding is critical to conservation and biodiversity research, yet public reference databases are incomplete. Existing barcode databases are biased toward cytochrome oxidase subunit I (COI) and frequently lack associated voucher specimens or geospatial metadata, which can hinder reliable species assignments. The emergence of metabarcoding approaches such as environmental DNA (eDNA) has necessitated multiple marker techniques combined with barcode reference databases backed by voucher specimens. Reference barcodes have traditionally been generated by Sanger sequencing, however sequencing multiple markers is costly for large numbers of specimens, requires multiple separate PCR reactions, and limits resulting sequences to targeted regions. High-throughput sequencing techniques such as genome skimming enable assembly of complete mitogenomes, which contain the most commonly used barcoding loci (e.g., COI, 12S, 16S), as well as nuclear ribosomal repeat regions (e.g., ITS1&2, 18S). We evaluated the feasibility of genome skimming to generate barcode references databases for marine fishes by assembling complete mitogenomes and nuclear ribosomal repeats. We tested genome skimming across a taxonomically diverse selection of 12 marine fish species from the collections of the National Museum of Natural History, Smithsonian Institution. We generated two sequencing libraries per species to test the impact of shearing method (enzymatic or mechanical), extraction method (kit-based or automated), and input DNA concentration. We produced complete mitogenomes for all non-chondrichthyans (11/12 species) and assembled nuclear ribosomal repeats (18S-ITS1-5.8S-ITS2-28S) for all taxa. The quality and completeness of mitogenome assemblies was not impacted by shearing method, extraction method or input DNA concentration. Our results reaffirm that genome skimming is an efficient and (at scale) cost-effective method to generate all mitochondrial and common nuclear DNA barcoding loci for multiple species simultaneously, which has great potential to scale for future projects and facilitate completing barcode reference databases for marine fishes.

Introduction

DNA barcoding has long been recognized as a critical component of biodiversity research (Hebert et al., 2003; Ratnasingham & Hebert, 2013; DeSalle & Goldstein, 2019; Adamowicz et al., 2019), but available barcode reference databases remain incomplete (Mugnai et al., 2021). More comprehensive regional reference datasets in global databases better support research goals and applications such as discovering new species (Carpenter, Williams & Santos, 2017; Hoban & Williams, 2020), matching larval specimens to known adults (Johnson et al., 2009; Hubert et al., 2010), and authenticating seafood labeling (Marko, Nance & Guynn, 2011; Silva & Hellberg, 2021). Traditionally, DNA barcoding efforts relied on Sanger sequencing of single mitochondrial markers, particularly cytochrome oxidase subunit I (COI) for metazoans. However, there is increasing utility for other mitochondrial genes and noncoding regions (e.g., 16S, 12S) as well as nuclear ribosomal genes that are present in tandem repeats (e.g., 18S-ITS1-5.8S-ITS2-28S) (Pochon et al., 2013; Berry et al., 2017; Alexander et al., 2020). To develop more complete DNA barcode databases, we evaluated a method of genome skimming that has potential to simultaneously recover multiple barcoding loci for many species.

DNA barcodes are essential resources, but the quality and utility of existing data is variable. For DNA barcodes to be of long-term value, they should be linked to physical voucher specimens in permanent natural history collections because voucher specimens allow for verification of identification and refinements in taxonomy (Schander & Willassen, 2005; Ward, Hanner & Hebert, 2009; but see Collins & Cruickshank, 2013). Another consideration for building barcode libraries stems from natural genetic variation in populations. For example, Hawaiian populations of widespread Indo-Pacific fishes are often genetically divergent and can comprise cryptic lineages (DiBattista et al., 2010; DiBattista et al., 2012; Bowen et al., 2013). Thus, the most valuable barcode sequences are derived from voucher specimens associated with precise geospatial metadata (geotags), which are unfortunately missing for most archived genomic datasets (Toczydlowski et al., 2021). Other attributes, such as color photographs of the specimen at the time of collection and detailed collection metadata, add to barcode value. Finally, to increase discoverability and data access, specimen and sequence metadata need to be linked through persistent digital identifiers across systems of record (Riginos et al., 2020). These best practices in data stewardship are necessary to support cross-domain cyberinfrastructure to enable transdisciplinary research, discovery and reuse of material samples and their derived data (Davies et al., 2021).

Efforts to characterize community biodiversity patterns through metabarcoding (Leray & Knowlton, 2015; Timmers et al., 2021) and environmental DNA (eDNA) surveys (Ficetola et al., 2008)—that rely on well-curated barcode databases to accurately assign sequences to taxonomy—have expanded dramatically (Ruppert, Kline & Rahman, 2019). In addition, approaches (such as eDNA) that are based on potentially fragmentary source material and/or those that target specific taxa are more precise with a multi-marker approach (Stat et al., 2017; West et al., 2020; Casey et al., 2021). Finally, targeting short hypervariable loci (e.g., Riaz et al., 2011; Miya et al., 2015) can be more compatible with read lengths produced by high-throughput sequencing (HTS) platforms. The availability of multiple genetic markers associated with a single voucher specimen also makes species identifications more consistent across studies where researchers may use different loci.

As high-throughput sequencing has become more accessible and cost-effective, genome skimming, which uses low-pass, shallow shotgun sequencing of whole genomes, has become practical (Trevisan et al., 2019). Genome skimming does not enrich samples for specific target loci, yet it is successful at recovering high-copy regions such as mitochondrial and plastid genomes as well as nuclear or cytosolic sequences like ribosomal DNA (Kane et al., 2012; Straub et al., 2012; Besnard et al., 2013; Malé et al., 2014; Ripma, Simpson & Hasenstab-Lehman, 2014; Dodsworth, 2015; Denver et al., 2016; Grandjean et al., 2017; Liu et al., 2020; Raupach et al., 2022). Genome skimming has great potential to fill DNA barcode reference databases because it generates sequence data for commonly used barcoding markers simultaneously (Coissac et al., 2016). This potential has been realized in a range of taxa from plants (Alsos et al., 2020) to arthropods (Grandjean et al., 2017; Raupach et al., 2022). Our work complements Therkildsen & Palumbi (2017), who used a similar approach to examine genetic variation in Atlantic Silversides and Margaryan et al. (2021), who developed a mitogenome barcode database for vertebrates of Denmark, and extends these studies by showing that ribosomal barcoding loci are also readily accessible using genome skimming. Despite previous applications of genome skimming, it has yet to be tested broadly to capture specimen-backed DNA barcodes for marine fishes.

Natural history collections hold valuable materials to support regional and taxon-specific barcode database development, allowing gaps to be filled without the need to collect new specimens. While many institutions voucher tissue samples and/or DNA extractions alongside collected specimens, sequences are frequently published for only a limited number of loci (e.g., COI for metazoans, ITS for fungi (Ratnasingham & Hebert, 2007)). In our study, which is part of an ongoing effort to complete the barcode reference database for Hawaiian marine fishes, we evaluated genome skimming as a method to rapidly and (when scaled up to massively parallel sequencing platforms) inexpensively capture all commonly used DNA barcoding loci for multiple samples and fish taxa simultaneously. Using genome skimming, we aimed to recover the complete mitochondrial genomes and ribosomal repeat regions of 12 taxonomically diverse species of marine fishes. For our test, we prepared and sequenced two libraries for each species (24 libraries total) from vouchered specimens in the National Museum of Natural History (NMNH) fish collection. We evaluated the quality of sequences and our ability to assemble complete mitogenomes and ribosomal repeats in the context of taxonomic diversity and shearing method, and across a range of DNA extraction methods and input DNA concentrations. Here we report the results of our test and discuss how to adapt this method for large-scale generation of specimen-backed DNA barcodes.

Materials & Methods

Sample selection

We selected samples from 12 species across a broad taxonomic distribution of fishes, including one chondrichthyan and 11 teleosts (Fig. 1). This work is a component of an effort to generate specimen-backed barcodes for all species of Hawaiian marine fishes (∼1,200 species; unpublished updated version of Mundy, 2005; Randall, 2007); thus, most specimens were Hawaiian species collected in Hawai‘i (6/12) or species that occur in Hawai‘i but that were collected elsewhere (3/12). We also included two western North Atlantic species: Brosme brosme (Cusk), which is a NOAA species of concern, and Gymnura altavela (Spiny Butterfly Ray), as a representative chondrichthyan. All samples were taken from existing DNA extracts in the National Museum of Natural History (NMNH) Biorepository, derived from specimens housed in the fish collection at NMNH (Table 1). Archived Biorepository DNA was originally extracted from tissues subsampled and preserved in the field at the time of specimen collection. Ten of the 12 specimens have live color photographs (Fig. 1). No mitogenomes or ribosomal repeats were available in GenBank for any of the species selected except Gymnura altavela, which was published during preparation of this manuscript (Kousteni et al., 2021). All selected Hawaiian species lacked regionally localized specimen-backed barcodes for at least one common fish barcoding locus (COI, 16S, 12S; Table S1).

Figure 1 Species included in this MiSeq-based pilot study.

(A) Gymnura altavela, Spiny Butterfly Ray, length unknown. (B) Gymnothorax fimbriatus, Fimbriated moray, USNM 395396, 850 mm TL. (C) Gymnothorax undulatus, Undulated moray, USNM 442319, 132 mm TL. (D) Saurida nebulosa, Clouded Lizardfish, USNM 442473, 56.2 mm SL. (E) Brosme brosme, Cusk, length unknown. (F) Myripristis vittata, Whitetip Soldierfish, USNM 411102, 120.1 mm SL. (G) Neoniphon sammara, Sammara Squirrelfish, USNM 442483, 130 mm SL. (H) Tylosurus crocodilus, Houndfish, USNM 442362, 13.6 mm SL. (I) Scomberoides lysan, Doublespotted Queenfish, USNM 442297, 22.3 mm SL. (J) Forcipiger flavissimus, Longnose Butterflyfish, USNM 411089, 129.1 mm SL. (K) Ostracion whitleyi, Whitley’s Boxfish, USNM 411029, 81.2 mm SL. (L) Canthigaster amboinensis, Ambon Toby, USNM 442417, 64 mm SL. All photographs except A and E are the individuals for which we sequenced the mitogenome. Photographs A and E by Donald D. Flescher, NOAA; photographs B, F, J, and K by Jeff Williams, NMNH; and photographs C, D, G, H, I, and L by Diane Pitassy NMNH.

Table 1 Summary of species and museum specimens included in this study.

Species in this and subsequent tables are arranged alphabetically by taxonomic order, family, and scientific name, with the chondrichthyan presented separately.

Scientific name	Order	Family	Extraction method	Estimated genome size (Gb)	USNM catalog number	Date collected	COI reference accession	
Gymnura altavela (Linnaeus, 1758)	Myliobatiformes	Gymnuridae	AutoGen	1.80c	433343	11 Sep. 2006	MH378654	
Gymnothorax fimbriatus (Bennett, 1832)	Anguilliformes	Muraenidae	BioSprint	2.31b	395396	15 Oct. 2008	MK658634	
Gymnothorax undulatus (Lacepède, 1803)	Anguilliformes	Muraenidae	AutoGen	2.31b	442319	26 May 2017	MG816692	
Saurida nebulosa Valenciennes, 1850	Aulopiformes	Synodontidae	AutoGen	1.53b	442473	1 Jun. 2017	MG816726	
Tylosurus crocodilus (Péron & Lesueur, 1821)	Beloniformes	Belonidae	AutoGen	1.00a	442362	28 May 2017	MG816741	
Myripristis vittata Valenciennes, 1831	Beryciformes	Holocentridae	BioSprint	0.90b	411102	16 Oct. 2008	MZ598162	
Neoniphon sammara (Forsskål, 1775)	Beryciformes	Holocentridae	AutoGen	0.80a	442483	31 May 2017	MG816708	
Brosme brosme (Ascanius, 1772)	Gadiformes	Lotidae	AutoGen	0.41a	433199	20 Apr. 2008	MH378533	
Scomberoides lysan (Forsskål, 1775)	Perciformes	Carangidae	AutoGen	0.73b	442297	25 May 2017	MG816730	
Forcipiger flavissimus Jordan & McGregor, 1898	Perciformes	Chaetodontidae	BioSprint	0.72a	411089	17 Oct. 2008	MK657435	
Ostracion whitleyi Fowler, 1931	Tetraodontiformes	Ostraciidae	BioSprint	0.98b	411029	15 Oct. 2008	MK658705	
Canthigaster amboinensis (Bleeker, 1864)	Tetraodontiformes	Tetraodontidae	AutoGen	0.41b	442417	30 May 2017	MG816661	
Notes.

a Genome size estimates were available for this exact species on NCBI and/or genomesize.com.

b Genome size estimates were calculated based on an average of available congeners or confamilials on NCBI and/or genomesize.com.

c Genome size estimate for this species was based on an average of members of Batoidea available on NCBI and/or genomesize.com.

DNA concentration and extractions

DNA extracts representing a range of concentrations (0.9–34.0 ng/µL) were retrieved from the NMNH Biorepository. We did not standardize concentrations prior to library preparation. To demonstrate that the two extraction methods commonly used at NMNH yield viable outcomes, we included four samples extracted with the Qiagen BioSprint DNA blood kit (Qiagen, Inc., Venlo, Netherlands) and eight samples extracted by an AutoGenPrep 965 automated DNA extraction robot (AutoGen, Holliston, MA, USA) following the manufacturer’s tissue protocols. These are standard DNA extraction technologies used for Sanger-based DNA barcoding, similar to those that have been used to generate the majority of available DNA extracts in existing collections.

Shearing method and library preparation

We prepared two libraries for each of the 12 fish species, one sheared enzymatically and the other sheared mechanically, for a total of 24 libraries. Input DNA for the mechanically sheared libraries was prepared using a Covaris ME220 sonicator (Covaris, Woburn, MA, USA), then libraries were constructed with the NEB Ultra II DNA library prep kit (New England Biolabs, Ipswich, MA, USA) according to the manufacturer’s protocols (with the exception noted below). We prepared enzymatically sheared libraries using the NEB Ultra II FS DNA library prep kit (New England Biolabs), which incorporates enzymatic shearing as part of the kit workflow. We targeted an insert size of approximately 200 bp and amplified libraries using six cycles of PCR according to the kit manufacturer’s chemistry and thermocycler settings. We used iTru y-yoke adapter stubs and iTru unique dual indices (Glenn et al., 2019) in place of NEB adapters and indices and tailored the amount of adapter based on DNA concentration following NEB guidelines. Individual libraries were quantified with a Qubit dsDNA HS assay (Thermo Fisher Scientific, Waltham, MA, USA) and run on a High Sensitivity D1000 ScreenTape (Agilent, Santa Clara, CA, USA) to assess library size in bp. Finally, libraries were pooled to equimolar amounts prior to sequencing.

During library preparation, our enzymatically sheared samples inadvertently sat at 4 °C following the end of the ligation period for an additional 45 min compared to those mechanically sheared. This gave the enzymatically sheared samples more time to ligate and likely impacted their ligation efficiency and subsequent library yield.

Sequencing

Libraries were split into two pools, and each pool was sequenced in a single run on the Illumina MiSeq (Illumina Inc., San Diego, CA, USA) using V3 chemistry at the Laboratories of Analytical Biology, NMNH. We limited the sequencing run length to 150 bp (paired end) to test scalability to higher-throughput platforms such as the Illumina NovaSeq 6000.

Assembly

We assessed two approaches to mitogenome assembly using Geneious Prime 2021.2.2 (https://www.geneious.com). First, we used the Map to Reference function and built-in Geneious mapper with the sensitivity set to “medium/low” and iterations set to “up to 10 times”, starting with published COI sequences (Table 1) for each of the 24 libraries. Resulting assemblies were inspected and trimmed at the ends (up to 50 bp) where coverage was low (<5X). Consensus sequences were generated from the assembly results and used as subsequent reference seeds and the Map to Reference step repeated until the assemblies stopped increasing in size and identical stretches of sequences were detected at the 5′ and 3′ ends. The second approach used a complete mitogenome from either a congeneric or confamilial taxon as the reference sequence, and Map to Reference, using the same parameters for a single set of up to 10 iterations. Assemblies of ribosomal repeat regions were conducted similarly, with reiterations using the Map to Reference function in Geneious, using ribosomal sequences from closely related taxa published in GenBank (Table S2). In addition to assembling mitogenomes, we constructed nuclear genome preassemblies using SPAdes 3.15.3 (assembly module only) on paired forward and reverse read libraries (Prjibelski et al., 2020), and filtered out preassembly contigs shorter than 200 bp.

Genome sequencing coverage estimation

We estimated species genome sizes (Table 1) based on data available in GenBank or the Animal Genome Size Database (Gregory, 2021). Where specific estimates were unavailable, we calculated an average genome size of congeners or closely related confamilials. Since no congener or confamilial genomes were available for G. altavela, we estimated genome size based on the average genome size for Batoidea. We then estimated sequence coverage (C) for each sample using the equation C = LN/G, where L was the sequencing read length, N was the number of reads, and G was the estimated haploid genome length.

Annotation

We annotated assembled mitogenomes using the MitoAnnotator tool from the MitoFish Mitochondrial Genome Database of Fish (Iwasaki et al., 2013). We manually annotated ribosomal repeat regions by aligning to complete ribosomal repeat regions for fishes in GenBank (Table 2). We did not annotate preassembly contigs.

Table 2 GenBank accession numbers for assembled mitogenomes and ribosomal repeat regions.

Species	Accession number (mitogenome)	Mitogeone length (bp)	Accession number (ribosomal repeat region)	DOI for Genome preassemblies and assembly statistics	
Gymnura altavela	OK104094	19,022a	MZ286332	10.5281/zenodo.5507151	
Gymnothorax fimbriatus	MZ297479	16,567	MZ286333	10.5281/zenodo.5507064	
Gymnothorax undulatus	MZ329992	16,566	MZ286339	10.5281/zenodo.5507172	
Saurida nebulosa	MZ329994	16,717	MZ286340	10.5281/zenodo.5507186	
Tylosurus crocodilus	MZ329993	16,533	MZ286342	10.5281/zenodo.5507182	
Myripristis vittata	MZ329989	16,520	MZ286336	10.5281/zenodo.5507128	
Neoniphon sammara	MZ329995	16,743	MZ286341	10.5281/zenodo.5507201	
Brosme brosme	MZ329990	16,483	MZ286337	10.5281/zenodo.5507143	
Scomberoides lysan	MZ329991	16,767	MZ286338	10.5281/zenodo.5507164	
Forcipiger flavissimus	MZ329988	16,600	MZ286335	10.5281/zenodo.5507111	
Ostracion whitleyi	MZ297480	16,461	MZ286334	10.5281/zenodo.5507077	
Canthigaster amboinensis	MZ188982	16,444	MZ188965	10.5281/zenodo.4753123	
Notes.

a Based on nearly-complete mitogenome assembly.

Phylogenetic analyses

To assess relationships and validate taxonomic identities, we performed phylogenetic analyses including all mitogenomes generated in this study and confamilial taxa with published mitogenomes available in the MitoFish database (52 species; Table S5). Due to the large number of species with available mitogenomes in the family Carangidae, we only used species in Seriola, Elegatis, and Decapterus, the available genera most closely related to our taxon Scomberoides lysan (Damerau, Freese & Hanel, 2018; Rabosky et al., 2018). We used sequences of all protein-coding genes (PCGs) and two rRNAs. Each PCG or rRNA was individually aligned using MAFFT v7.505 (Katoh & Standley, 2013) and then concatenated to a single final alignment. We used PartitionFinder2 to assess the partitioning of models of molecular evolution (Guindon et al., 2010; Lanfear et al., 2012; Lanfear et al., 2017). We partitioned the alignment by gene and, for PCGs, by codon position, then ran PartitionFinder2 using the “greedy” algorithm with branch lengths specified as unlinked to test the models supported in MrBayes. We conducted a Bayesian phylogenetic reconstruction using MrBayes v3.2.7 (Ronquist et al., 2012), running four independent searches of six chains for 22 million generations, saving trees every 1,000 generation and discarding the first 15% as burn-in. We verified convergence of MCMC runs and model parameters using TRACER v1.7.2 (Rambaut et al., 2018). We conducted a maximum likelihood (ML) phylogenetic reconstruction with the partitioned alignment using RAxML v8.2.12 (Stamatakis, 2014) and specified 1,000 bootstrap replicates to assess node support. Resulting trees were rooted to the two Gymnura species and plotted using ggtree (Yu et al., 2017) and phytools (Revell, 2012) in R v4.1.2 (R Core Team, 2020).

Data availability

All voucher and material sample properties can be found in GEOME, the Genomic Observatories Metadatabase (Riginos et al., 2020), under the expedition NMFS_FISHES_MiSeqPilot_01 (https://n2t.net/ark:/21547/EEV2). We deposited BioSample records, annotated mitogenome and ribosomal repeat assemblies, and raw reads in GenBank (BioProject Accession: PRJNA720393). Code and procedures used to perform phylogenetic analyses are available on GitHub (https://github.com/hawaii-barcoding-initiative/mitogenome_tree).

Results

DNA concentration

Total input DNA for library preparation ranged from 4.6 to 170 ng. Final libraries ranged from 0.16 to 3.34 ng/µL in concentration, with mechanically and enzymatically sheared libraries averaging 0.71 ± 0.67 ng/µL (mean ± sd) and 1.72 ± 0.94 ng/µL, respectively. The average total library size ranged from 318 to 392 bp, with mechanically- and enzymatically sheared libraries averaging 345 ± 16 bp and 373 ± 18 bp, respectively. A summary of library quantification results can be found in Table 3.

Table 3 Library quantification and sequencing results; values shown are for both shearing methods (mechanical; enzymatic).

Species	Input DNA for library preparation (ng)	Average library size (bp)	Final library concentration (ng/µL)	Total raw reads	Calculated genome coverage	Reads mapped to mitogenome	Percent reads mapped	Avg. mitogenome coverage	
Gymnura altavela	170	318; 326	2.50; 1.98	2,193,690; 2,224,022	0.18; 0.19	201; 1,141	0.01; 0.05	1.6; 8.9	
Gymnothorax fimbriatus	78	353; 370	0.498; 1.31	1,522,912; 1,809,632	0.10; 0.12	2,336; 2,647	0.15; 0.15	20.7; 23.0	
Gymnothorax undulatus	51	356; 379	0.984; 2.82	2,146,906; 5,168,856	0.14; 0.34	984; 2,245	0.05; 0.04	8.7; 19.5	
Saurida nebulosa	27.6	353; 391	0.382; 1.87	2,120,606; 3,174,282	0.21; 0.31	5,290; 5,603	0.25; 0.18	47.1; 48.7	
Tylosurus crocodilus	4.6	380; 390	0.156; 0.27	463,424; 2,451,640	0.07; 0.37	1,065; 5,507	0.23; 0.22	9.4; 48.6	
Myripristis vittata	25.1	337; 354	0.352; 1.42	1,290,468; 2,342,102	0.21; 0.39	754; 1,615	0.06; 0.07	6.7; 13.8	
Neoniphon sammara	17.1	352; 375	0.286; 0.876	2,276,566; 4,265,046	0.43; 0.80	2,169; 3,957	0.10; 0.09	19.3; 34.6	
Brosme brosme	41	334; 392	0.366; 1.79	1,027,598; 1,635,836	0.37; 0.69	3,321; 5,148	0.32; 0.31	29.4; 45.1	
Scomberoides lysan	33.9	340; 378	0.344; 1.30	2,621,818; 4,818,598	0.54; 0.99	7,249; 12,324	0.28; 0.26	64.2; 107.9	
Forcipiger flavissimus	109	351; 378	1.06; 2.96	1,993,702; 2,116,356	0.41; 0.44	1,193; 1,311	0.06; 0.06	10.5; 11.1	
Ostracion whitleyi	86.5	340; 371	1.32; 3.34	2,054,668; 2,473,712	0.31; 0.38	2,369; 3,069	0.12; 0.12	20.596; 27.089	
Canthigaster amboinensis	19.1	331; 371	0.224; 0.678	1,880,384; 2,868,978	0.68; 1.04	6,070; 8,672	0.32; 0.30	53.132; 76.469	

Sequence reads and genome coverage

We recovered 0.46 to 5.2 million reads (2.5 ± 1.1 million) per library. AutoGen and Qiagen extractions performed comparably (2.6 ± 1.3 million reads for AutoGen vs. 2.0 ± 0.4 million for Qiagen). Enzymatic shearing yielded more reads per library than mechanical shearing (2.9 ± 1.1 million reads for enzymatic vs. 1.8 ± 0.6 million reads for mechanical). Sequence duplication rates varied from 0.7–6.5% per sample. Based on estimated genome sizes, these read counts equate to 0.07× to 1.04× genome coverage, with enzymatic shearing (0.50 ± 0.30×) averaging higher than mechanical shearing (0.30 ± 0.19×). A summary of sequencing results across libraries is presented in Table 3.

Assembly and sequence coverage

We readily assembled and annotated complete mitochondrial genomes for the 11 teleosts (see Table 2 for assembled mitogenome accession numbers). Assembled sequences were identical whether we started from a small seed (COI) or mapped to a complete mitochondrial reference genome derived from a congeneric or confamilial taxon. We did not recover a complete mitogenome from Gymnura altavela (Spiny Butterfly Ray), but assembled large sections of it (e.g., ∼12,000 bp including COI; ∼3,000 bp including 16S). During the course of this work, a complete mitochondrial genome was published for G. altavela (MT274571) based on a specimen from Greece (Kousteni et al., 2021). This allowed us to improve our assembly, resulting in a mitochondrial genome with a short gap in COI and a second gap in the D-loop. Fortunately, the gap spanned the published COI sequence for our specimen (USNM 433343; MH378654), allowing us to use 24 bases from that sequence to fill the missing space. As a result, we ultimately derived a nearly-complete mitochondrial genome for the Spiny Butterfly Ray (19,022 bp in our assembly as compared to 19,472 bp in MT274571).

Mitogenome coverage of the 22 successful assemblies ranged from 7× to 108× (34 ± 26×; Table 3). The Gymnura altavela libraries had a comparable number of reads to other species in our study, but coverage of the mitogenome was low for unknown reasons (11.2× with both libraries combined). Across all libraries, assembled mitogenome reads comprised 0.05% to 0.32% (0.17 ± 0.1%) of the total raw reads generated per specimen.

Using Geneious Map to Reference, we assembled and annotated ribosomal repeat regions (18S-ITS1-5.8S-ITS2-28S) for all 12 taxa by using 18S or 28S reference seeds (see Table 2 for assembled ribosomal repeat accession numbers).

Genome preassemblies generated by SPAdes (>200 bp) were uploaded to Zenodo (along with basic assembly statistics) and assigned persistent identifiers (Table 2). As expected, the preassemblies were limited, with a small fraction of contigs exceeding 1 kb in length. Nevertheless, preassembly contigs that correspond to the complete or nearly complete mitochondrial genomes and the ribosomal repeat regions were recovered for 7 and 8, respectively, of the 12 species in our study.

Mitogenome organization and structure

Mitogenomes for all species were arranged similarly, with some minor length variations, particularly in the control region (see Fig. 2 for example assembly of Canthigaster amboinensis; see Fig. S1 for all mitogenome assemblies). We detected no mitochondrial gene rearrangements among the 12 species we investigated. All species had 36 genes comprising 13 protein-coding genes (PCGs) and 23 tRNAs, with two rRNAs and the control region. In all cases, the majority strand encoded 12 PCGs, 15 tRNAs, both rRNAs, and the control region. The remaining eight tRNAs and a single PCG were encoded on the minority strand. GC content ranged from 43.1% (Neoniphon sammara) to 52.1% (Gymnothorax fimbriatus) (mean: 45.5 ± 2.3%).

Figure 2 Assembled and annotated mitogenome of Canthigaster amboinensis, Ambon Toby, USNM 442417, 64 mm SL.

Photograph by Diane Pitassy, NMNH.

Phylogenetic analyses

Both ML and Bayesian methods produced identical topologies (Fig. 3), with the single exception of different branching order within the genus Ostracion (see Figs. S2 and S3 for raw ML and Bayesian trees respectively, including complete node support values). All specimens sequenced for this study were recovered within their respective taxonomic groups, and branching order among families matched that of the family-level backbone tree published in Rabosky et al. (2018). Node support in the ML tree (bootstrap value) was more variable than in the Bayesian tree (posterior probability). The ML tree had many strongly supported nodes (70–100% bootstrap support), but two with weak support (<20%). The Bayesian tree was strongly supported throughout, with most nodes having >95% posterior probability.

Figure 3 Results of phylogenetic analysis of 52 fish mitogenomes.

Tree shown is the result of the Bayesian analysis confirming that 12 focal taxa (shown in red) are correctly placed among confamilials in corresponding families. Node support values <95% are shown for nodes at family- and genus-level splits. Bayesian posterior probability is as labeled, and ML bootstrap support is indicated by the color of the node symbols. Unlabeled family- and genus-level nodes had 100% posterior probability and bootstrap support.

Discussion

Our results show that genome skimming by shallow shotgun sequencing is an efficient method for generating mitogenomes and ribosomal repeats of marine fishes. The methods are robust for a broad range of taxa, extraction types, shearing methods, and DNA concentrations. Both kit-based (Qiagen) and automated (AutoGen) extractions resulted in high quality sequence libraries, which indicates that this method can leverage existing DNA extractions housed in museum collections that were prepared for other purposes (e.g., single-marker Sanger sequencing).

As noted in Methods, our enzymatically sheared samples were held at 4 °C following the end of the ligation period for an additional 45 min compared to those mechanically sheared. This likely impacted their ligation efficiency and subsequent library yield. As a result, we cannot confirm that differences in final library yield nor differences in read counts resulted directly from the shearing method used. Although libraries were pooled in equimolar ratios, these values were calculated using total dsDNA (ng/µL) as measured by Qubit and TapeStation fragment size (bp). A more accurate method would be to quantify individual libraries with qPCR, thus measuring DNA that can be sequenced, rather than total DNA. Regardless of these differences, we demonstrated that enzymatic shearing can be an effective method for genome skimming; enzymatic shearing is also less expensive (∼$4 less/library; Tables S3 and S4), less labor intensive, and requires less specialized laboratory equipment.

We assembled mitogenomes with as few as half a million reads but had more consistent success with 2–3 million reads/library, which resulted in an average of 34 × coverage of the mitogenome.

Mitogenome assemblies used only 0.05% to 0.32% of the total raw sequence reads. The majority of unassembled reads were nuclear (e.g., chromosomal) and cytosolic (e.g., ribosomal RNA) sequences. The most common barcoding markers for fishes are mitochondrial: COI (Leray et al., 2013), 16S rRNA (Berry et al., 2017), and 12S rRNA (Miya et al., 2015). However, primer sets designed to amplify other taxa or communities often target nuclear ribosomal loci such as the 18S rRNA and/or internal transcribed spacers (ITS1/2) (marine eukaryotes: Pochon et al., 2013; scleractinian corals: Alexander et al., 2020). We successfully recovered complete ribosomal repeat regions (18S-ITS1-5.8S-ITS2-28S) from all of our sequence libraries, illustrating that our approach has applications beyond mitogenome assembly. Importantly, we recovered sequences for the most commonly used barcoding loci for all targeted taxa in a single pass. We provided raw sequence data in the NCBI Sequence Read Archive under BioProject PRJNA720393 because there are likely additional sequences of interest to other researchers. In addition, we constructed genome preassemblies for each sample, which are also available (Table 2).

Our phylogenetic analyses of concatenated protein-coding genes and rRNAs recovered a topology consistent across tree-building methods (Fig. 3) and that comports with recent higher-level fish phylogenies (e.g., Rabosky et al., 2018). Notably, in the combined tree the two nodes most weakly supported by ML both had 100% Bayesian posterior probabilities. These discrepancies may be due to how the two approaches partition molecular evolution models. RAxML supports partitioning but only allows a single model across the alignment, whereas MrBayes allows models to vary across partitions (e.g., genes & codon positions). Overall, this phylogenetic analysis helps validate both the species identity of each voucher specimen and the quality of genome-skimming derived mitogenome assemblies.

To test whether our methods are applicable across fish diversity, we included one chondrichthyan, the Spiny Butterfly Ray Gymnura altavela. Despite high success across teleosts, we did not recover a complete mitogenome for the chondrichthyan. The G. altavela libraries had read counts comparable to bony fish libraries, but mitogenome coverage was low and initial assemblies had gaps. Two potential causes of low coverage in Gymnura include exogenous (non-target) DNA and a greater number of shorter-than-expected DNA fragments in the sequencing libraries. During preparation of this manuscript, a complete mitochondrial genome was published from a specimen from Greece (Kousteni et al., 2021), and although it is ∼3% diverged from our mitochondrial sequences, we used it to improve our assembly such that it included complete loci other than the D-loop. Gaps in the control region are relatively common in mitochondrial genome assemblies, particularly among rays (Poortvliet et al., 2015; Hinojosa-Alvarez et al., 2015). This region often contains tandem repeats that present difficulty to bioinformatic assemblers (White et al., 2018) and have been attributed to heteroplasmy in other taxa (Mundy, Winchell & Woodruff, 1996). However, despite the D-loop gap in the complete mitogenome assembly of G. altavela, we still recovered targeted mitochondrial barcoding loci (COI, 12S, 16S). In future studies, we will sequence additional sharks, rays, and chimaeras to further explore laboratory and bioinformatic approaches for generating chondrichthyan mitogenomes.

We used the MiSeq platform to test shearing methods, compare extraction types and DNA concentrations, and to assess sequencing reads and coverage necessary to generate mitogenomes and ribosomal repeats across a broad taxonomic selection of fishes. To further our goal of completing barcode reference databases (for mitochondrial and ribosomal genes) for all species of Hawaiian fishes, we will sequence future genome skimming runs on an Illumina NovaSeq. The NovaSeq platform produces higher read output than MiSeq and therefore supports increased multiplexing of samples, allowing us to pool 384 samples (species) in a single sequencing run. This will reduce sequencing costs from ∼$145 per sample on the MiSeq to ∼$16 on the NovaSeq, while also increasing the average yield from 2.5 million reads to 13 million per sample. The increased multiplexing capability of the NovaSeq brings the total cost (library preparation, quantitation, and sequencing) from ∼$161 per sample on the MiSeq to ∼$31 per sample, which will facilitate economical and rapid generation of complete mitogenomes and ribosomal repeats (encompassing all major barcoding loci) (see Tables S3 and S4). Preliminary data (not reported here) from a NovaSeq run of 384 species show that our methods for mitogenome and ribosomal repeat recovery via genome skimming can be scaled to the higher-throughput platform. In this study, we employed manual assembly methods using Geneious Prime, whereas future assemblies will employ an automated bioinformatic pipeline to enable production of multilocus DNA barcode sequences at scale.

We enhanced the reference value of our derived genetic data through persistent digital identifiers. Raw reads and assembled sequences are linked through NCBI accessions (BioProject, BioSample, SRA, and nucleotide) to museum voucher specimens, as well as to derived tissues and DNA extracts at NMNH. Further, to ensure that data derived from and associated with these biomaterials can easily be accessed and reused, we cross linked NCBI and GEOME records through Archival Resource Key (ARK) identifiers (Kunze, 2021). Such best practices in data stewardship and the use of persistent identifiers across systems of record will facilitate cross-domain cyberinfrastructure and enable transdisciplinary research, discovery, and reuse of material samples and their derived data (Davies et al., 2021).

Conclusions

Our study shows that genome skimming is an efficient and cost effective method that will allow a shift in the DNA barcoding workflow from sequencing targeted loci in individual specimens to generating complete suites of barcode markers for many taxa in a single sequencing run. The methods we employed enable use of genetic samples housed in natural history collections to rapidly generate specimen-based, regionally localized DNA barcode reference data. This work has important implications for several large US-based initiatives: NOAA ‘omics (Goodwin et al., 2021), NMNH Ocean DNA Initiative (https://www.smithsonianmag.com/blogs/national-museum-of-natural-history/2021/07/07/meet-reef-expert-collecting-environmental-time-capsules/), and the US Ocean Biocode (Meyer et al., 2021), each of which involves explicit aims to provide complete DNA barcode reference databases based on voucher specimens. Techniques and methods developed here are applicable to taxa and regions beyond marine fishes and the Hawaiian Islands. Comprehensive voucher-based reference databases are necessary to advance sequence-based detection, censusing, and monitoring of marine communities in the face of global change.

Supplemental Information

Table S1 Summary of published barcode sequences for selected species

The sequences from GenBank were accessed March 2022. Sequences were defined as having valid associated specimens when the/specimen_voucher qualifier was present and conformed to valid NCBI BioCollections standard institution and (where included) collection codes.

Click here for additional data file.

Table S2 Reference sequences used to assemble ribosomal repeat regions. All reference sequences cover 18S-ITS1-5.8S-ITS3-28S

Click here for additional data file.

Table S3 Cost summary for this study’s pilot genome skimming experiment (12 libraries prepped with enzymatic shearing, and 12 libraries with mechanical shearing; sequenced on MiSeq)

Click here for additional data file.

Table S4 Cost summary for scaled-up genome skimming (384 libraries on NovSeq)

Click here for additional data file.

Table S5 List of species and GenBank accession numbers used for phylogenetic reconstruction

Click here for additional data file.

Figure S1 All assembled mitogenomes from this study

(A)Gymnura altavela. (B) Gymnothorax fimbriatus. (C) Gymnothorax undulatus. (D) Saurida nebulosa. (E) Tylosurus crocodilus. (F) Myripristis vittata. (G) Neoniphon sammara. (H) Brosme brosme. (I) Scomberoides lysan. (J) Forcipiger flavissimus. (K) Ostracion whitleyi. (L) Canthigaster amboinensis.

Click here for additional data file.

Figure S2 Raw maximum likelihood tree with bootstrap support for all nodes, rooted to the two Gymnura spp

Click here for additional data file.

Figure S3 Raw Bayesian tree with node support (posterior probability percentage) for all nodes, rooted to the two Gymnura spp

Click here for additional data file.

Samples from French Polynesia were acquired under a collaborative Centre de Recherche Insulaire et Observatoire de l’Environnement (CRIOBE) and Smithsonian Institution National Museum of Natural History (NMNH) initiative to survey the marine fishes of French Polynesia, including the Mo‘orea Biocode Project. We thank NMNH and CRIOBE, in particular Jeffrey Williams (NMNH) and Serge Planes (CRIOBE), as the collectors and photographers of the samples. Specimens collected from Hawai‘i were acquired under the MarineGEO Hawai‘i 2017 project to survey the fishes of Kāne‘ohe Bay. We thank the Smithsonian Conservation Biology Institute, NMNH, and the Hawai‘i Institute of Marine Biology, in particular Mary Hagedorn, Lynne R. Parenti, Diane Pitassy, Zeehan Jaafar, Kassi S. Cole, and Kiril Vinnikov, as the collectors of the samples. Photographs of Hawaiian fishes were provided by Diane Pitassy. The technical support and advanced computing resources from University of Hawaii Information Technology Services – Cyberinfrastructure are gratefully acknowledged. Genetic benchwork and sequencing was completed at the Smithsonian NMNH Laboratories of Analytical Biology (LAB). At NMNH we thank Carole Baldwin, Daniel DiMichele, Chris Huddleston, Lynne R. Parenti, Diane Pitassy, Niamh Redmond, Makiri Sei, Lee Weigt, Jeff Williams, and Herman Wirshing for their support. We thank Cheryl Morrison, Andreas Zwick, and an anonymous reviewer for kind, thoughtful, and constructive comments on an earlier version of this paper. This is contribution #1893 from the Hawai‘i Institute of Marine Biology, #11535 from the School of Ocean and Earth Science and Technology at the University of Hawai’i, and #2022_20 from NOAA’s West Hawai’i Integrated Ecosystem Assessment Program.

Additional Information and Declarations

Competing Interests

Author Contributions

Data Availability

The authors declare there are no competing interests.

Mykle L. Hoban conceived and designed the experiments, performed the experiments, analyzed the data, prepared figures and/or tables, authored or reviewed drafts of the article, and approved the final draft.

Jonathan Whitney conceived and designed the experiments, performed the experiments, analyzed the data, authored or reviewed drafts of the article, and approved the final draft.

Allen G. Collins conceived and designed the experiments, performed the experiments, analyzed the data, prepared figures and/or tables, authored or reviewed drafts of the article, and approved the final draft.

Christopher Meyer conceived and designed the experiments, analyzed the data, authored or reviewed drafts of the article, and approved the final draft.

Katherine R. Murphy performed the experiments, analyzed the data, prepared figures and/or tables, authored or reviewed drafts of the article, and approved the final draft.

Abigail J. Reft analyzed the data, authored or reviewed drafts of the article, and approved the final draft.

Katherine E. Bemis conceived and designed the experiments, analyzed the data, prepared figures and/or tables, authored or reviewed drafts of the article, and approved the final draft.

The following information was supplied regarding data availability:

All sequences, assemblies, and short reads are available at GenBank: PRJNA720393.

All sample metadata is tracked in GEOME under the expedition ID NMFS_FISHES_ MiSeqPilot_01 at https://n2t.net/ark:/21547/EEV2.

The code and alignments used to generate and plot phylogenetic trees are available at GitHub: https://github.com/hawaii-barcoding-initiative/mitogenome_tree.

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
