# Peer review of "Skimming for barcodes: rapid production of mitochondrial genome and nuclear ribosomal repeat reference markers through shallow shotgun sequencing"

_PeerJ, doi:10.7717/peerj.13790_

## Round 0.1 · original submission · Minor Revisions

Dear Dr. Hoban and co-authors,

I have received three independent reviews of your study - two accept and one minor, well done! Please note the comments from Reviewer 1 and annotated comments from Reviewer 3.

I'd like to thank all three reviewers for their time and care in assessing this work.

Overall, the reviewers have provided you with excellent suggestions on how to improve the manuscript, and I be looking forward to receiving your revised manuscript along with a point-by-point response to their comments.

With warm regards,
Xavier

Reviewer 1 ·

Basic reporting

I believe that the indtroduction will benefit if the authors gave it a little more structure. Maybe in a time line sense i.e. , first mention traditional barcoding COI and Sanger sequencing ( third praragraph in the manuscript), then the importance of barcode and metadata, followed by other types of barcoding like eDNA, then the disadvantages and finishing with the aim of this study.

Experimental design

no comment

Validity of the findings

I was looking for a phylogentic tree, but there is none, I think it is important to add a phylogenetic tree with other previously sequenced sister species to furhter validate the authors findings. Please also provide the alingement used for the tree inference

Additional comments

The article provides a novel and cheaper methodology to optimeze the power of sequencing and use it to contribute to the biological databases and collections. It will improve with just a small extra analysis that will not consume a lot of exta time and resources.

·

Basic reporting

The article is written in clear English and is technically correct. Appropriate literature is cited. The figures and tables are well constructed, high quality, and provide relevant information. The results follow stated objectives and the overall hypothesis that genome skimming could be utilized to obtain full mitochondrial genomes and ribosomal RNA genes simultaneously.

Experimental design

The authors have described the knowledge gap they are targeting with their research well. High technical standards were followed in both lab methods as well as bioinformatics, and all are described in sufficient detail.

Validity of the findings

A difference in procedures followed between mechanical and enzymatic shearing makes the direct comparison of these methods obscure. The authors point out this potential problem, but this comparison is weakened.

Additional comments

This is a clearly and concisely written paper that explores several variables that contribute to successful genome skimming to obtain barcode markers across museum vouchered fish taxa. The results suggest that the methods work under varying (and realistic) DNA concentrations, extraction and shearing methods.

In the abstract, it may be more logical to discuss the aspects being tested in the order in which they occur in the process (DNA extraction, DNA input concentration, then shearing), as was done in the Introduction.

·

Basic reporting

Well-written in clear, unambiguous English. Introduction and background information are relevant and well-presented, making a strong case for the method presented. The manuscript structure conforms with PeerJ and general scientific publication standards. The two figures are well-done and relevant to illustrate taxonomic diversity and mt-genome annotation. Raw data, annotated assemblies and specimen metadata are all clearly linked to and accessible. Supplementary Tables S3 and S4 could do with a bit more context / explanation of what they present. If I understood S3 correctly, I'm not sure that all numbers in there are correct (no cost difference between enzymatic and Covaris shearing).

Experimental design

Original research that is well within scope of the journal. Research questions (e.g., “Can we generate multi-gene markers from collection specimens with genome skimming?”, “Is there a difference in DNA shearing method applied?”) are well defined and relevant, advancing our understanding of how to generate reference sequences for, e.g., eDNA studies. The investigation is sufficiently rigorous, with the main caveat being that multiple variables (DNA extraction method, DNA concentrations, DNA fragmentation method) were changed at the same time, which makes it impossible to correlate a particular change with the result (e.g., read coverage or recovery of the mt-genome). However, no significant differences in results were detected in any case – “it always worked, no matter what”. The methods are described in sufficient detail (additional info on flow cell type and read de-duplication would be helpful).

Validity of the findings

All underlying data have been detailed and are accessible, even at the review stage. The data are robust and sound. Replication of the genome skimming approach is explicitly justified by extending it to include ribosomal data. The conclusions are clear, supported by the data and answering the original research questions. The study provides an outlook on scaling up the approach, pointing to data for 384 samples not presented in this manuscript. However, this is not a major point of this manuscript and acceptable (the provided bioproject seems to include not only the data presented in this study, but also these additional unpublished data).

Additional comments

This manuscript makes a very compelling case for generating DNA reference sequences for molecular species identifications that are backed by verifiable voucher specimens, have all relevant specimen data included and are not limited to one particular molecular marker. These are exactly the drawbacks of existing “reference” sequences, and reading the abstract I thought “Yes, finally a paper that is to the point!”. Well done!
In terms of experimental design, I wish the study would have had more samples and only changed one variable (parameter) at a time to enable correlations between the variable and the outcome. This would be relevant when looking at performance and optimisation of workflows in more detail (which isn’t the topic of this manuscript, it aims at demonstrating the principle). As this study seems to have used only high-quality DNA extracts (from presumably frozen tissue samples), the outcome has always been successful and there is no apparent need for or difference between different approaches. When working with more degraded DNA (older specimens; formalin-fixed; stored at room temperature) and scaling up, this will be come more relevant – maybe the topic of a future paper.
I’ve placed various comments directly in the PDF file for consideration or information, but none of them should preclude this manuscript from getting published. It’s a good paper, and I’m looking forward to being able to cite it.

---

## Round 0.2 · accepted · Accept

Dear Dr Hoban and co-authors,

I am pleased to accept your revised manuscript for publication in PeerJ. Thank you for this great scientific contribution!

With warm regards,
Xavier

Reviewer 1 ·

Basic reporting

The authors have addressed all the introduction comets

Experimental design

Have no comments

Validity of the findings

no comment